# Digital interventions for common mental disorders in low- and middle-income countries: A systematic review and meta-analysis

Eirini Karyotaki[1,2,3] (iD), Clara Miguel[1,2,3], Olga M. Panagiotopoulou[1,2],
Mathias Harrer[4,5], Nadine Seward[6], Marit Sijbrandij[1,2,3], Ricardo Araya[6],
Vikram Patel[7,8] (iD) and Pim Cuijpers[1,2,3] (iD)

[1]Department of Clinical, Neuro-, and Developmental Psychology, Vrije Universiteit Amsterdam, Amsterdam, The Netherlands; [2]WHO Collaborating Centre for Research and Dissemination of Psychological Interventions, Vrije Universiteit Amsterdam, Amsterdam, The Netherlands; [3]Amsterdam Public Health Research Institute, Amsterdam, The Netherlands; [4]Psychology & Digital Mental Health Care, Department of Health Sciences, Technical University Munich, Munich, Germany; [5]Department of Clinical Psychology & Psychotherapy, Friedrich-Alexander-University Erlangen-Nuremberg, Erlangen, Germany; [6]Center for Global Mental Health and Primary Care Research, Health Service and Population Research, Institute of Psychiatry, Psychology and Neuroscience, King's College London, London, UK; [7]Department of Global Health and Social Medicine, Harvard Medical School, Boston, MA, USA and [8]Department of Global Health and Population, Harvard T.H. Chan School of Public Health, Harvard University, Boston, MA, USA

## Review

**Keywords:**
digital interventions; depression; anxiety; low- and middle-income countries; global health

**Corresponding author:**
Eirini Karyotaki;
Email: e.karyotaki@vu.nl

## Abstract

**Background:** In low-resource settings, e-mental health may substantially increase access to evidence-based interventions for common mental disorders. We conducted a systematic literature search to identify randomised trials examining the effects of digital interventions with or without therapeutic guidance compared to control conditions in individuals with anxiety and/or depression symptoms in low- and middle-income countries (LMICs).

**Methods:** The main outcome was the reduction in symptoms at the post-test. Secondary outcomes included improvements in quality of life and longer-term effects (≥20 weeks post-randomisation). The effect size Hedges' $g$ was calculated using the random effects model.

**Results:** A total of 21 studies (23 comparisons) with 5.296 participants were included. Digital interventions were more effective than controls in reducing symptoms of common mental disorders at the post-test ($g = -0.89$, 95% confidence interval [CI] $-1.26$ to $-0.52$, $p < 0.001$; NNT = 2.91). These significant effects were confirmed when examining depressive ($g = -0.77$, 95% CI $-1.11$; $-0.44$) and anxiety symptoms separately ($g = -1.02$, 95% CI $-1.53$ to $-0.52$) and across all other sensitivity analyses. Digital interventions also resulted in a small but significant effect in improving quality of life ($g = 0.32$, 95% CI 0.19 to 0.45) at the post-test. Over the longer term, the effects were smaller but remained significant for all examined outcomes. Heterogeneity was moderate to high in all analyses. Subgroup and meta-regression analyses did not result in significant outcomes in any of the examined variables (e.g., guided vs. unguided interventions).

**Conclusions:** Digital interventions, with or without guidance, may effectively bridge the gap between treatment supply and demand in LMICs. Nevertheless, more studies are needed to draw firm conclusions regarding the magnitude of the effects of digital interventions.

## Impact statement

The burden of common mental disorders, such as depression and anxiety, is devastating worldwide and especially high in low- and middle-income countries (LMICs), where mental health care resources are scarce. Scalable low-intensity psychological interventions may provide significant benefits in addressing the unmet mental health needs of individuals in LMICs. Over the last decades, digital interventions have shown promising effects in reducing symptoms of common mental disorders in high-income countries, and it has been suggested that such interventions may be beneficial for LMICs as well. Although the research in this field is relatively recent, the number of trials focussing on the effects of digital interventions in LMICs is booming, calling for an aggregated synthesis of the existing findings. Our systematic review and meta-analysis aimed to provide the latest evidence on the efficacy of digital interventions in reducing symptoms of anxiety and depression and improving the quality of life among individuals in LMICs. Digital interventions were significantly better in reducing symptoms of depression anxiety compared to control conditions at post-treatment and over the longer term. Significant improvements were also found in the quality-of-life outcomes. We also demonstrated that the effectiveness of digital interventions did not differ concerning several subgroup analyses like

intervention format, delivery platform, age groups and adaptation to the local situation. Although more research is needed to draw firm conclusions regarding the examined subgroup analyses, these findings are encouraging for the upscaling of digital interventions in LMICs especially considering that these interventions could be delivered without therapeutic guidance and still lead to significant outcomes. Overall, our findings testify to the generalisability of the effectiveness of digital psychological interventions for depression and anxiety beyond high-income countries. This innovative delivery strategy for psychological treatment should be scaled up to supplement existing healthcare resources.

## Introduction

Common mental disorders, like depression and anxiety, affect hundreds of millions of people (Steel et al., 2014) and are associated with disability, morbidity and even premature mortality (Reddy, 2010). The global burden of these disorders is devastating globally and particularly high in low- and middle-income countries (LMICs; Herrman et al., 2022; WHO, 2022). Although psychotherapy, pharmacotherapy, and their combination can effectively manage the symptoms of depression and anxiety (Cuijpers et al., 2020), their availability in LMICs is limited due to scarce healthcare resources. Insufficient access to evidence-based treatment results in chronicity and poorer disease prognosis, causing avoidable suffering. Therefore, research efforts in global mental health have focussed on improving access to psychological interventions in low-resource settings (Patel et al., 2018).

Using digital interventions, evidence-based treatments may be more widely accessible at lower costs. Besides cost reduction, such interventions can overcome many other treatment barriers since they do not exclusively rely on face-to-face sessions and mental health specialists (Andersson et al., 2019). Available options include digital interventions with/without therapeutic guidance. These interventions can be delivered in various ways, for example, using web platforms, mobile apps and chatbots. In recent years, the growing interest in alternative intervention delivery modes, particularly in high-income countries (HICs), has led to a research boom on digital interventions for depression and anxiety. These interventions have shown promising short- and long-term results in HICs (Karyotaki et al., 2021; Pauley et al., 2021), prompting questions about their potential for filtering down to LMICs and their efficacy within those contexts.

Considering the rapidly increasing penetration rate of digital technologies in LMICs, digital interventions may be an effective approach to fill the gap between treatment supply and demand (Fairburn and Patel, 2017; Karyotaki et al., 2017; Naslund et al., 2017). However, there still exists a digital divide, particularly among the lowest socioeconomic status groups and vulnerable populations. Barriers, including access to the internet/a personal smartphone, power outages and low literacy, also present challenges in implementing digital interventions in LMICs. These factors highlight the critical need to provide an overview of the available research evidence, specifically from lower-income settings. Such an overview will enable a comprehensive examination of the effectiveness of digital interventions in these contexts, shedding light on their potential impact and identifying the gaps in our current knowledge.

Thus far, the evidence of the effectiveness of digital interventions primarily comes from Western HICs, and little is known about the effects of such therapeutic approaches in LMICs. Only one narrative review has summarised preliminary evidence on using digital technology to treat and prevent mental disorders in LMICs (Naslund et al., 2017). Given the recent booming of clinical trials in this field, it is imperative to synthesise at a meta-analytic level their effects in LMICs. Thus, we aim to provide the current state of the art of digital interventions in LMICs by examining their overall effects in reducing symptoms of depression and/or anxiety as compared to control conditions at the post-test. Secondary analyses included quality of life and longer-term outcomes.

## Methods

The present systematic review and meta-analysis was reported in accordance with the PRISMA statement (Page et al., 2021).

### Search strategy

For this study, we ran a systematic literature search using free and index terms indicative of depression, anxiety and digital interventions in PubMed, PsycINFO and Embase from database inception through 25 February 2022 (the full search strings for PubMed can be found in the Supplementary Material). Next to these targeted searches, we performed an additional search in an existing broader living meta-analytic database on psychological treatments of depression (Cuijpers et al., 2020; doi:10.17605/OSF.IO/825C6), of which supplemental materials and information are available on the website of the project (www.metapsy.org). To develop this living database, we continuously search four major bibliographical databases (i.e., PubMed, PsycINFO, Embase and the Cochrane Library) by combining index and free terms indicative of depression and psychotherapies with filters for randomised controlled trials. These searches are conducted every 4 months, and the current meta-analysis includes references up to 1 January 2022. Furthermore, we checked the references of earlier meta-analyses on psychological treatments for depression and anxiety. Two independent researchers screened all records, and all papers that could meet inclusion criteria were retrieved as full text. The two independent researchers also decided to include or exclude a study in the database, and disagreements were resolved through discussion.

### Selection of studies

In the current meta-analysis, we included randomised trials conducted in LMICs where a digital intervention for people with anxiety and/or depression was compared to a control condition. To determine whether a country was considered an LMIC, we used the World Bank classification data for the year in which the trial was published. Digital interventions were defined as psychological interventions delivered via the internet through a web platform or a mobile application. Given that digital interventions are novel in LMICs, we included all delivery formats in the present meta-analysis (i.e., guided, self-guided and blended) to give a broad overview of what exists in this context. Depression and anxiety were determined as meeting diagnostic criteria for a depressive or anxiety disorder or scoring above a validated cut-off on a self-report measure. Control conditions had to be inactive, such as waitlist, care as usual, or attention placebo. No age-related exclusion criteria were applied since we wanted to examine the effects of these

interventions across the lifespan. Nevertheless, results were also reported separately for adults and adolescents to examine potential differences between these two groups.

### Data extraction and risk of bias assessment

We extracted general characteristics of the participants (i.e., type of disorder, the diagnostic method at inclusion, recruitment method, the proportion of women and mean age), characteristics of the treatment (i.e., type of digital intervention, the guidance received along the intervention, number of sessions or modules and cultural adaptation) and characteristics of the studies (i.e., type of control group, the country where the trial was conducted, income level of the country and year of publication).

Included studies were assessed using the Cochrane Risk of Bias Tool 2 (Sterne et al., 2019). This tool evaluates possible sources of bias in five domains: 1) biases arising from the randomisation process, 2) deviations from the intended interventions, 3) missing outcome data, 4) biases in the measurement of the outcome and 5) selective reporting of the result. Each previous domain was rated as (a) low risk, (b) some concerns or (c) high risk of bias. Based on the scores of the domains, each trial received an overall risk of bias score of low risk (when all domains were scored as low risk), some concerns (when at least one domain is rated as some concerns, but none of the domains has a high-risk score), and high risk (when at least one domain is scored as high risk).

### Outcome measures

We assessed the effectiveness of digital interventions on common mental health symptoms (i.e., depression and anxiety) and quality of life outcomes. For each comparison between a digital intervention and a control condition, the effect size (Hedges' $g$) indicating the difference between the two groups at the post-test was calculated to adjust for small sample sizes (Hedges and Olkin, 2014). Effect sizes were calculated by subtracting the intervention group's mean score from the control group's mean score and dividing the result by the pooled standard deviation. Additionally, we calculated effect sizes at long-term follow-ups (≥20 weeks since randomisation). When a trial reported data at multiple follow-up points, we took the longest follow-up from the trial.

### Meta-analyses

We examined the effects of digital interventions by running the main meta-analysis on common mental health symptoms, which combined depression and anxiety measures. We also ran analyses to determine the effects on the two outcomes separately. In addition, we performed a meta-analysis on quality-of-life outcomes. To account for dependency between effect sizes (e.g., when multiple instruments were used to measure the same outcome), we pooled all outcomes within a comparison before calculating the overall effect. To pool effect sizes within the studies, we assumed an intra-study correlation coefficient of $\rho = 0.5$. This method of effect size pooling was used as the primary analysis method.

Additionally, we conducted sensitivity analyses using other methods for pooling by running: 1) a generic inverse-variance model and assuming all effect sizes as independent, 2) excluding outliers (effect sizes whose 95% confidence interval [CI] does not overlap with the CI of the pooled effect) and calculating the overall effect when only the smallest and highest effect size within a study is considered. Examining and addressing outliers is essential to ensure

the validity of the meta-analytic model's assumptions and to examine sources of heterogeneity. Outlying studies with extremely large effect sizes or a small number of highly influential studies can distort the pooled effect estimate. To assess the potential impact of outliers, metapsyTools incorporates two sensitivity analyses. The first approach employs a basic outlier removal strategy, known as the non-overlapping confidence intervals method, while the second utilises an influence analysis using the leave-one-out method, as outlined by Viechtbauer and Cheung (2010). These sensitivity analyses serve the purpose of checking the robustness of the model by examining the rationale for excluding potential outliers and influential cases. Such rigorous examination of outliers contributes to maintaining the integrity and reliability of the meta-analytic findings. Moreover, we examined publication bias through Egger's test of the intercept (Egger et al., 1997) and Duval and Tweedie's trim and fill procedure (Duval and Tweedie, 2000).

A random-effects model was used in all the analyses, using the restricted maximum likelihood method. For models not fitted using RVE, we applied the Knapp–Hartung method to obtain robust confidence intervals and significance tests of the overall effect (IntHout et al., 2014). We examined the heterogeneity of effect sizes by calculating the $I^2$-statistic and its 95% CI. This statistic indicates the presence of heterogeneity in percentages, with 25% as low, 50% as moderate and 75% as high heterogeneity (Higgins et al., 2003). For the three-level models, we calculated a multilevel extension of $I^2$, which describes the amount of total variability attributable to heterogeneity within studies (level 2) and heterogeneity between studies (level 3) (Cheung, 2014; Harrer et al., 2021). Further, we calculated prediction intervals (PIs), which indicate the range in which the true effect size of 95% of all populations will fall (Borenstein et al., 2017).

In our main meta-analysis of common mental health symptoms, we conducted a series of subgroup analyses that examined differences in effect sizes based on the type of technology used, type of intervention, format, income level of the country, age group, cultural adaptation of the intervention, method of recruitment, diagnosis of the included sample, type of diagnosis ascertainment at inclusion, and target group, and type of control condition. All subgroup analyses were conducted using a mixed-effects model. We pooled only subgroups with five or more studies.

All the analyses were conducted in RStudio 4.2.0 using the 'metapsyTools' package, which imports functionalities of the 'meta' (Balduzzi et al., 2019), 'metafor' (Viechtbauer, 2010), and 'dmetar' (Harrer et al., 2021) packages.

## Results

### Selection and inclusion of studies

A total of 17,204 records were identified through the targeted searches. After removing duplicates, 9,419 titles and abstracts were screened, of which 948 papers were retrieved for full-text selection. Next, 53 full texts on digital interventions were examined from the living Metapsy database and three more studies were identified through reference tracking. All full texts were screened against our current eligibility criteria resulting in 21 RCTs for inclusion in the present meta-analysis (Figure 1; Mogoaşe et al., 2013; Tulbure et al., 2015; Yeung et al., 2017; Arjadi et al., 2018; Ciuca et al., 2018; Karbasi and Haratian, 2018; Yeung et al., 2018; Moeini et al., 2019; Jannati et al., 2020; Salamanca-Sanabria et al., 2020; Wang et al., 2020; Araya et al., 2021; Ghosh et al., 2023; Heim et al., 2021; Latif

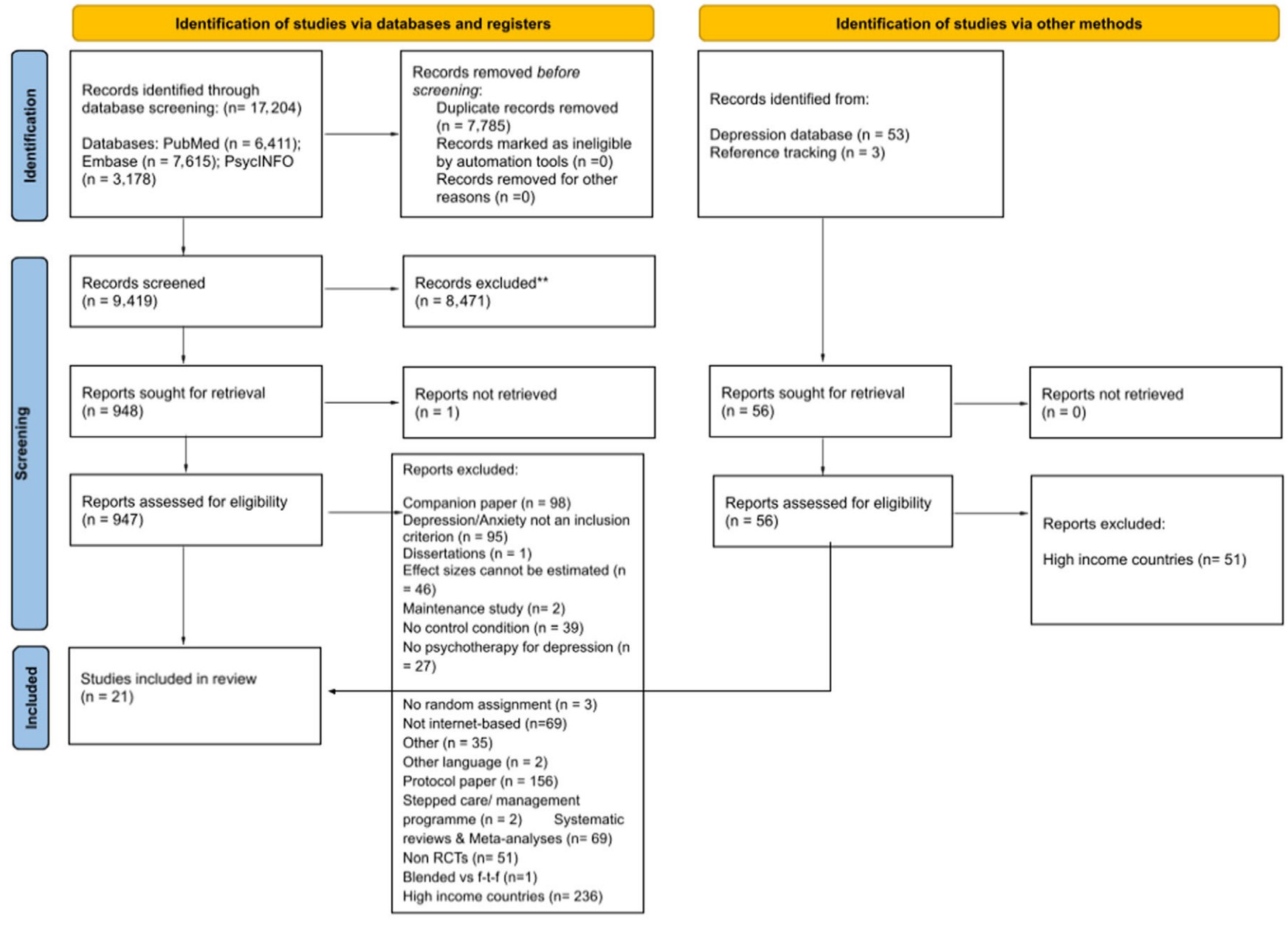

**Figure 1.** Flow chart of the studies selection process.

et al., 2021; Newman et al., 2021; Sun et al., 2021; Cuijpers et al., 2022; Zhao et al., 2022).

### Characteristics of included studies

The characteristics of the included studies are presented in Table 1. A total of 5,296 participants (2,701 in the interventions and 2,595 in the control groups) were included in 21 trials consisting of 23 comparisons. Fifteen trials focussed primarily on depressive symptoms, five on anxiety and one on depression and/or anxiety. Most of the studies employed cut-off scores on a self-report scale, while six performed a diagnostic interview to include participants. Further, most of the 23 comparisons examined a web-based ($n = 17$) cognitive behavioural therapy ($n = 13$) and had some form of guidance ($n = 15$). Across the 21 included studies, the most prevalent control condition was the waiting list ($n = 12$), and the most prevalent target group was adults in general ($n = 9$). Post-treatment outcomes were assessed in a period ranging from 1 week to 4 months, while the respective time of assessment for follow-up outcomes ranged from 5 to 9 months post-randomisation. The included trials were conducted in 10 lower- and upper-middle countries (i.e., Brazil, China, Colombia, India, Indonesia, Iran, Lebanon, Pakistan, Peru and Romania) according to the World Bank classification at the time when the studies were published.

### Risk of bias assessment

Overall, the risk of bias ranged from some concerns to high risk across all included studies. There were some concerns for most of the studies related to bias arising from the randomisation process and selection of the reported results, primarily due to the lack of information regarding the randomisation/allocation process and pre-registrations of outcome measures. Most of the studies were at low risk, related to deviations from the intended interventions and bias in the measurement of the outcome. However, bias was prevalent in many of the included studies due to missing outcome data. Since the overall judgement of the studies was either some concerns or high risk, we could not perform a sensitivity analysis including only low risk of bias studies. Figure 2a presents the overall risk of bias across studies, and Figure 2b shows the risk of bias assessment per study.

### Effects of digital interventions on symptoms of common mental disorders

Table 2 presents the effects of digital interventions compared to controls. At the post-treatment, digital interventions had a large effect in reducing symptoms of common mental disorders ($g = -0.89$, 95% CI $-1.26$ to $-0.52$, $p < 0.001$; NNT = 2.91; Figure 3). Heterogeneity was high and significant ($I^2 = 87.75\%$; 95% CI 82.93 to 91.21; PI = $-2.5$ to 0.72). There was an indication of publication bias (Egger's test $p = 0.002$), resulting in a much smaller but significant adjusted effect of $g = -0.49$. All sensitivity analyses yielded similar effects with high heterogeneity and wide confidence intervals. However, when outliers were excluded, the effect dropped to $g = -0.67$ (95% CI $-0.82$ to $-0.51$), and the heterogeneity was reduced to moderate $I^2 = 68.26$ (95% CI 48.28 to 80.53; PI $-1.23$ to $-0.11$). We found no evidence of a difference between all examined subgroup analyses ($p > 0.05$) except for the type of therapy, with cognitive behaviour therapy showing significantly larger effects ($g = -1.37$) than other types of psychotherapies ($g = -0.53$)

($p = 0.007$). The results of subgroup analyses are presented in Table 3.

Further, we conducted separate analyses on depression and anxiety outcomes. Regarding depressive symptoms, digital interventions resulted in an overall effect of $g = -0.77$ (95% CI $-1.11$ to $-0.44$; $p < 0.001$; NNT = 3.34). Heterogeneity was high ($I^2 = 85.32$, 95% CI 78.62 to 89.92; PI = $-2.13$ to 0.59). When two outliers were removed, the effect dropped to $g = -0.57$, and heterogeneity was moderate $I^2 = 66.62$ (95% CI 45.27 to 79.65; PI = $-1.08$; $-0.07$). There was an indication of publication bias (Egger's test $p = 0.02$), and Trim and fill resulted in an effect size of $g = -0.47$ after adjusting for six missing studies. A bit larger effect was found when examining the effects of digital interventions on anxiety symptoms at the post-test. Digital interventions resulted in a large effect of $g = -1.02$ (95% CI $-1.53$ to $-0.52$; $p < 0.001$; NNT = 2.58) in reducing anxiety symptoms. Heterogeneity was high ($I^2 = 83.90$, 95% CI 74.39 to 89.88; PI = $-2.71$ to 0.66) but dropped to moderate when excluding two outliers ($g = -0.74$, 95% CI $-0.96$; $-0.51$; $I^2 = 67.47$, 95% CI 40.46 to 82.23). The overall effect on anxiety symptoms dropped to $g = -0.62$ after adjusting for four missing studies, while the Egger's test was significant ($p = 0.01$). Similar outcomes were found in sensitivity analyses in which all outcome measures were treated separately, and the study with the largest/ smaller effect size was excluded. Table 2 presents all results of sensitivity analyses.

### Secondary outcomes (i.e., quality of life and longer-term effects)

At the post-test, digital interventions had a small effect on improving quality of life ($g = 0.32$, 95% CI 0.19; 0.45, $p < 0.001$; NNT = 8.12) based on seven studies reporting on this outcome. Heterogeneity was moderate $I^2 = 44.09$ (95% CI 0 to 76.48; PI = 0.04 to 0.61). No outliers were identified. The effect slightly changed to $g = 0.28$ after adjusting for two missing studies in the Trim and Fill procedure. However, Egger's test was non-significant ($p = 0.19$).

Eight studies reported long-term outcomes ≥20 weeks post-randomisation). Overall, digital interventions had a small effect compared to controls in reducing symptoms of common mental disorders over the longer term ($g = -0.31$, 95% CI $-0.49$ to $-0.12$; $p = 0.006$; NNT = 8.39). Heterogeneity was moderate $I^2 = 69.23$ (95% CI 35.78 to 85.26; PI = $-0.77$; 0.16). There were no outliers or indications of publication bias. Similar small but significant effects were found when examining depression ($g = -0.30$) and anxiety symptoms separately ($g = -0.45$) and across all other sensitivity analyses. Finally, digital interventions had a small effect in improving quality of life over the longer term ($g = 0.35$, 95% CI 0.12 to 0.58; NNT = 7.40) with moderate heterogeneity of $I^2 = 76.46$ (95% CI 47.3 to 89.49; PI $-0.23$ to 0.94). No outliers or indications of publication bias were identified. Table 2 presents all results of the secondary analyses.

### Discussion

In this systematic review, we examined the effects of digital interventions with or without guidance compared to control conditions in individuals with symptoms of common mental disorders in LMICs. At post-test, we found that digital interventions resulted in moderate to large effects on depression and/or anxiety symptoms, while there was a small but significant effect on quality of life. Over the longer term, the favourable effects of digital interventions were still significant, albeit smaller in magnitude. Heterogeneity

**Table 1.** Studies characteristics

| Study | Conditions | N | Platform | Sessions n | Criteria | Recruitment | Target group | Outcome | Assessment | Country | Income | Cultural adaptation |
|---|---|---|---|---|---|---|---|---|---|---|---|---|
| Araya et al., 2021 (trial a) | BAT | 440 | Mobile | 18 | PHQ-9 ≥ 10 | Other | General medical | Response (50% PHQ-9); | 3; 6 m | Brazil | Upper-middle | Yes |
| | CAU | 440 | | | | | | QOL (EQ5D-3L) | | | | |
| Araya et al., 2021 (trial b) | BAT | 217 | Mobile | 18 | PHQ-9 ≥ 10 | Other | General medical | Response (50% PHQ-9); | 3; 6 m | Peru | Upper-middle | Yes |
| | CAU | 215 | | | | | | QOL (EQ5D-3L) | | | | |
| Arjadi et al., 2018 | BAT | 159 | Web | 8 | Mood disorder | Community | Adults | DS (PHQ-9); QOL (WHO-QOL-BREF) | 10 w | Indonesia | Lower-middle | Yes |
| | Other | 154 | | | | | | | | | | |
| Ciuca et al., 2018 | gCBT | 36 | Web | 16 | Panic disorder | Community | Adults | AS (PDSS-SR); DS (PHQ-9) | 3 m | Romania | Upper-middle | NS |
| | sgCBT | 37 | | | | | | | | | | |
| | WL | 38 | | | | | | | | | | |
| Cuijpers et al., 2022 (trial a) | BAT | 331 | Web | 5 | PHQ-9 > 10 | Community | Adults | DS (PHQ-9); AS (GAD-7); | 2; 5 m | Lebanon | Lower-middle | Yes |
| | Other | 349 | | | | | | QOL (WHO-5) | | | | |
| Cuijpers et al., 2022 (trial b) | BAT | 283 | Web | 5 | PHQ-9 > 10 | Community | Other | DS (PHQ-9); AS (GAD-7); | 2; 5 m | Lebanon | Lower-middle | Yes |
| | Other | 286 | | | | | | QOL (WHO-5) | | | | |
| Ghosh et al., 2023 | CBT | 204 | Web | 6 | 5 ≥ PHQ-9 ≤ 19 | Community | Young adults | DS (PHQ-9); AS (GAD-7) | NR | India | Lower-middle | NS |
| | WL | 205 | | | | | | | | | | |
| Guo, 2020 | Other | 150 | Mobile | 12 | CES-D ≥ 16 | Other | General Medical | DS (CES-D, PHQ-9); | 3; 9 m | China | Upper-middle | Yes |
| | WL | 150 | | | | | | QOL (WHOQOL-HIV BREF) | | | | |
| Heim et al., 2021 | BAT | 67 | Mobile | 5 | PHQ-9 ≥ 10 | Community | Adults | DS (PHQ-9); AS (GAD-7); | 2; 5 m | Lebanon | Lower-middle | Yes |
| | Other | 71 | | | | | | QOL (WHO-5) | | | | |
| Jannati et al., 2020 | CBT | 39 | Mobile | 8 | EPDS ≥13 | Community | Other | DS (EPDS) | 2 m | Iran | Lower-middle | NS |
| | WL | 39 | | | | | | | | | | |
| Karbasi and Haratian, 2018 | CBT | 15 | Web | 7 | Anxiety disorder | Clinical | Adolescents | AS (SCARED) | 4 m | Iran | Lower-middle | NS |
| | CAU | 15 | | | | | | | | | | |
| Latif et al., 2021 | CBT | 20 | Web | 7 | HADS ≥8 | Clinical | Adults | DS (HADS depression); | 3 m | Pakistan | Lower-middle | Yes |
| | WL | 19 | | | | | | AS (HADS anxiety) | | | | |

**Table 1.** (*Continued*)

| Study | Conditions | N | Platform | Sessions n | Criteria | Recruitment | Target group | Outcome | Assessment | Country | Income | Cultural adaptation |
|---|---|---|---|---|---|---|---|---|---|---|---|---|
| Moeini et al., 2019 | CBT | 64 | Web | 8 | 10 ≥ CES-D ≤ 45 | Community | Adolescents | DS (CES-D) | 3; 6 m | Iran | Lower-middle | NS |
| | CAU | 64 | | | | | | | | | | |
| Mogoaşe et al., 2013 | 3rd | 21 | Web | 7 | BDI-II ≥ 12 | Community | Adults | DS (BDI-II) | 1 w | Romania | Upper-middle | NS |
| | WL | 21 | | | | | | | | | | |
| Newman et al., 2021 | CBT | 117 | Web | 40 | GAD-Q-IV ≥ 5.7 | Community | College students | AS (GAD-Q-IV); DS (DASS-D) | 3 m | India | Lower-middle | No |
| | WL | 105 | | | | | | | | | | |
| Salamanca-Sanabria et al., 2020 | CBT | 107 | Web | 7 | 10 ≥ PHQ-9 ≤ 19 | Community | Young adults | DS (PHQ-9); AS (GAD-7) | 7 w | Colombia | Upper-middle | Yes |
| | WL | 107 | | | | | | | | | | |
| Sun et al., 2021 | 3rd | 84 | Mobile | 8 | EPDS >9 or | Other | Other | DS (EPDS); AS (GAD-7) | 2; 8 m | China | Upper-middle | NS |
| | Other | 84 | | | PHQ-9 > 4 | | | | | | | |
| Tulbure et al., 2015 | CBT | 38 | Web | 9 | SAD or | Community | Adults | AS (LSAS-SR, SPIN, SIAS, SPSQ); | 11 w | Romania | Upper-middle | NS |
| | WL | 38 | | | Subclinical SAD | | | DS (BDI-II) | | | | |
| Wang et al., 2020 | gCBT | 70 | Web | 8 | SAD | Community | Adults | AS (SIAS; SPS) | 2 | China | Upper-middle | NS |
| | sgCBT | 70 | Web | | | | | | | | | |
| | WL | 70 | | | | | | | | | | |
| Yeung et al., 2017 | CBT | 37 | Web | 5 | Sign. depression symptoms | Clinical | Adults | DS (CES-D) | 5 w | China | Upper-middle | No |
| | WL | 38 | | | | | | | | | | |
| Zhao et al., 2022 | 3rd | 95 | Web | 6 | BDI-II ≥ 14 | Community | College students | DS (BDI-II) | 2 m | China | Upper-middle | Yes |
| | WL | 87 | | | | | | | | | | |

Abbreviations: 3rd, third wave cognitive behavioural therapy; AS, anxiety severity; BAT, behavioural activation therapy; BDI-II, Beck Depression Inventory-II; CAU, care-as-usual control group; CBT, cognitive behavioural therapy; CES-D, Center for Epidemiologic Studies Depression Scale; DASS-D, Depression Anxiety Stress Scales Short Form-Depression Subscale; DS, depression severity; EPDS, Edinburgh Postnatal Depression Scale; EQ5D-3L, European Quality of Life 5 Dimensions 3 Level Version; GAD-7, Generalised Anxiety Disorder 7; GAD-Q-IV, Generalised Anxiety Disorder Questionnaire for DSM-IV; gCBT, guided cognitive behavioural therapy; HADS, Hospital Anxiety and Depression Scale; LSAS-SR, Liebowitz Social Anxiety Scale-Self Report; m, month; NR, not reported; NS, non-specified; PDSS-SR, Panic Disorder Severity Scale – Self Report; PHQ-9, Patient Health Questionnaire-9; QOL, quality of life; SAD, social anxiety disorder; SCARED, Screen for Child Anxiety Related Emotional Disorders; sgCBT, self-guided cognitive behavioural therapy; SIAS, Social Interaction and Anxiety Scale; Sign, significant; SPIN, Social Phobia Inventory; SPS, Social Phobia Scale; SPSQ, Social Phobia Screening Questionnaire; w, week; WHO-5, The World Health Organisation – Five Well-Being Index; WHO-QOL-BREF, World Health Organisation Quality-of-Life Scale (abbreviated version); WHOQOL-HIV BREF, World Health Organisation Quality-of-Life Scale for people living with HIV (human immunodeficiency virus); WL, waiting list control group.

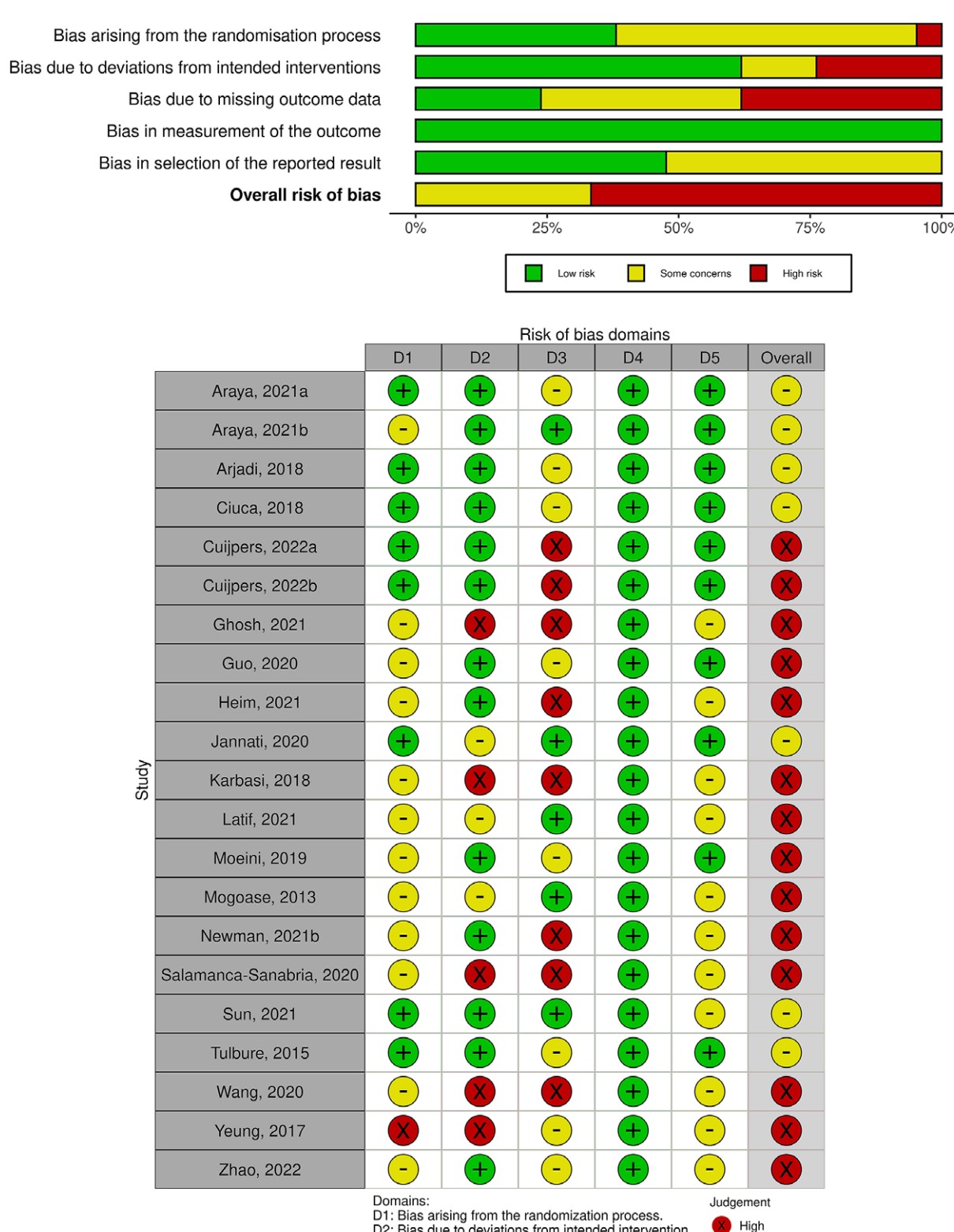

**Figure 2.** (a) Overall risk of bias across studies; (b) Risk of bias per each study.

was moderate to high across all comparisons, and there were indications of publication bias, suggesting that the present findings, although promising, should still be interpreted cautiously. Finally, we found no evidence of a difference between most of the examined subgroup analyses, including the intervention format, delivery platform, age groups and adaptation to the local situation. However, CBT programmes appeared to be more effective than other types of therapy.

**Table 2.** Effects of digital interventions on common mental disorders in LMICs

| Outcome | k | g | CI | p | $I^2$ | CI | PI | NNT |
|---|---|---|---|---|---|---|---|---|
| **Post-treatment** | | | | | | | | |
| Overall effects | 23 | −0.89 | (−1.26; −0.52) | <0.001 | 87.75 | (82.93; 91.21) | (−2.5; 0.72) | 2.91 |
| One ES/study (highest) | 21 | −0.96 | (−1.38; −0.55) | <0.001 | 88.48 | (83.78; 91.81) | (−2.69; 0.77) | 2.72 |
| One ES/study (lowest) | 21 | −0.81 | (−1.23; −0.4) | <0.001 | 87.22 | (81.82; 91.01) | (−2.54; 0.91) | 3.18 |
| Outliers removed | 18 | −0.67 | (−0.82; −0.51) | <0.001 | 68.26 | (48.28; 80.53) | (−1.23; −0.11) | 3.82 |
| Effect sizes separately | 40 | −0.88 | (−1.11; −0.65) | <0.001 | 83.58 | (78.45; 87.49) | (−2.12; 0.35) | 2.94 |
| Depression symptoms | 20 | −0.77 | (−1.11; −0.44) | <0.001 | 85.32 | (78.62; 89.92) | (−2.13; 0.59) | 3.34 |
| Anxiety symptoms | 14 | −1.02 | (−1.53; −0.52) | <0.001 | 83.90 | (74.39; 89.88) | (−2.71; 0.66) | 2.58 |
| Quality of life | 7 | 0.32 | (0.19; 0.45) | <0.001 | 44.09 | (0; 76.48) | (0.04; 0.61) | 8.12 |
| **Follow-up** | | | | | | | | |
| Overall effects | 8 | −0.31 | (−0.49; −0.12) | 0.006 | 69.23 | (35.78; 85.26) | (−0.77; 0.16) | 8.39 |
| One ES/study (highest) | 8 | −0.33 | (−0.53; −0.13) | 0.005 | 68.81 | (34.77; 85.09) | (−0.82; 0.16) | 7.87 |
| One ES/study (lowest) | 8 | −0.27 | (−0.46; −0.09) | 0.009 | 62.19 | (18.43; 82.47) | (−0.71; 0.16) | 9.68 |
| Effect sizes separately | 13 | −0.35 | (−0.48; −0.21) | <0.001 | 62.51 | (31.69; 79.42) | (−0.73; 0.04) | 7.40 |
| Depression symptoms | 8 | −0.30 | (−0.51; −0.09) | 0.012 | 69.91 | (37.42; 85.53) | (−0.81; 0.2) | 8.68 |
| Anxiety symptoms | 4 | −0.45 | (−0.71; −0.2) | 0.011 | 18.12 | (0; 87.46) | (−0.77; −0.14) | 5.72 |
| Quality of life | 6 | 0.35 | (0.12; 0.58) | 0.011 | 76.46 | (47.3; 89.49) | (−0.23; 0.94) | 7.40 |

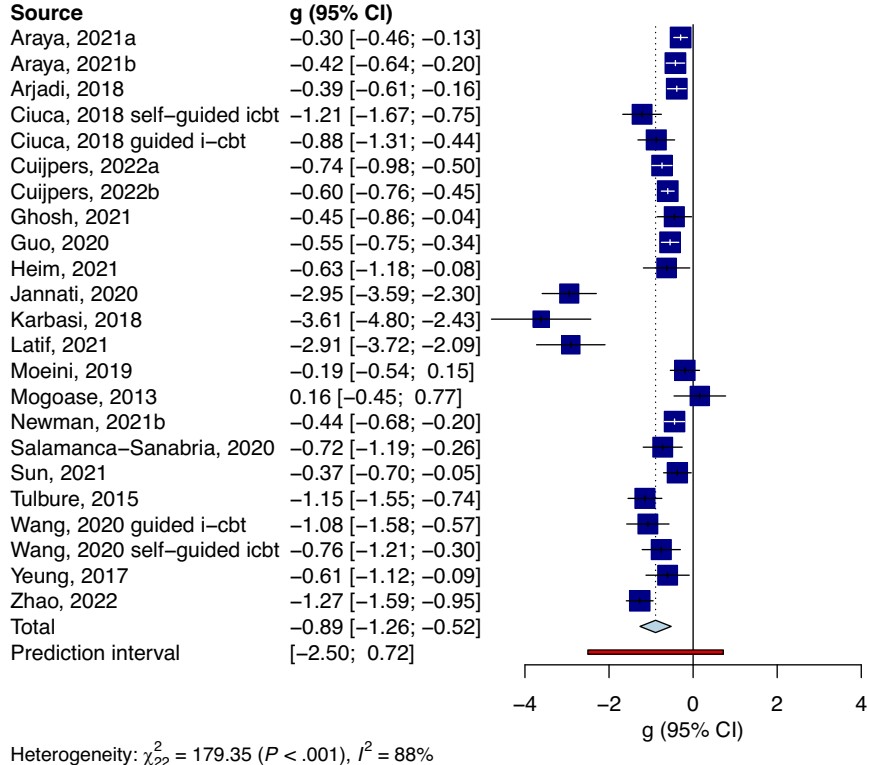

**Figure 3.** Forest plot of the main outcome.

A previous meta-analysis on this topic focussed only on clinical diagnosis and pooled together outcomes of several mental disorders, including post-traumatic stress disorder and psychosis (Fu et al., 2020). This is the first meta-analysis that pools together outcomes of trials on digital interventions in LMICs for anxiety and depression across various severities and regardless of a clinical diagnosis. The present effect sizes for depression ($g = −0.77$) and anxiety ($g = −1.02$) are slightly higher compared to those observed

**Table 3.** Subgroup analyses of digital interventions on symptoms of common mental disorders in LMICs at post-treatment

| Subgroup | $n_{comp}$ | g | CI | $I^2$ | CI | NNT | p |
|---|---|---|---|---|---|---|---|
| **Main platform** | | | | | | | |
| App-based | 6 | −0.84 | (−1.89; 0.21) | 92.0 | (85.3; 95.6) | 3.07 | 0.875 |
| Web-based | 17 | −0.91 | (−1.34; −0.48) | 85.0 | (77.4; 90.1) | 2.85 | |
| **Format** | | | | | | | |
| Guided | 15 | −0.90 | (−1.38; −0.43) | 86.0 | (78.4; 90.9) | 2.88 | 0.928 |
| Unguided | 8 | −0.87 | (−1.63; −0.11) | 90.6 | (83.8; 94.5) | 2.97 | |
| **Type of intervention** | | | | | | | |
| Cognitive behavioural therapy | 13 | −1.37 | (−2.01; −0.73) | 90.5 | (85.6; 93.7) | 2.04 | 0.007 |
| Other psychotherapies | 10 | −0.53 | (−0.77; −0.3) | 77.8 | (59.4; 87.9) | 4.84 | |
| **Type of control** | | | | | | | |
| Waiting list | 14 | −1.14 | (−1.65; −0.64) | 88.9 | (83.1; 92.7) | 2.34 | 0.194 |
| Other control conditions | 8 | −0.66 | (−1.32; −0.01) | 81.4 | (65.8; 89.9) | 3.88 | |
| **Outcome domain** | | | | | | | |
| Depression | 19 | −0.78 | (−1.14; −0.41) | 87.3 | (81.7; 91.3) | 3.30 | 0.229 |
| Anxiety | 4 | −1.56 | (−3.54; 0.43) | 84.7 | (61.8; 93.9) | 1.87 | |
| **Sample analysed** | | | | | | | |
| Completers | 12 | −0.66 | (−1.1; −0.22) | 78.5 | (63; 87.5) | 3.88 | 0.179 |
| Intention-to-treat | 10 | −1.15 | (−1.84; −0.46) | 92.5 | (88.2; 95.2) | 2.33 | |
| **Income-based on World Bank** | | | | | | | |
| Upper middle | 13 | −0.70 | (−0.94; −0.46) | 80.1 | (66.8; 88.1) | 3.66 | 0.210 |
| Lower middle | 10 | −1.22 | (−2.13; −0.32) | 92.4 | (88.1; 95.2) | 2.22 | |
| **Cultural adaptation** | | | | | | | |
| Yes | 10 | −0.79 | (−1.28; −0.3) | 86.7 | (77.5; 92.2) | 3.25 | 0.585 |
| No/non specified | 13 | −0.98 | (−1.59; −0.37) | 88.9 | (82.9; 92.8) | 2.67 | |
| **Age group** | | | | | | | |
| Adults | 18 | −0.89 | (−1.27; −0.51) | 88.4 | (83.2; 92) | 2.91 | 0.869 |
| Adolescents/young adults | 5 | −0.99 | (−2.66; 0.68) | 87.0 | (72; 94) | 2.64 | |
| **Recruitment** | | | | | | | |
| Community | 16 | −0.90 | (−1.28; −0.52) | 86.5 | (79.7; 91.1) | 2.88 | 0.620 |
| Other | 7 | −1.17 | (−2.41; 0.07) | 91.1 | (84.2; 95) | 2.30 | |
| **Diagnosis** | | | | | | | |
| Cut-off scores | 17 | −0.80 | (−1.21; −0.38) | 88.4 | (83; 92.1) | 3.22 | 0.376 |
| Clinical | 6 | −1.21 | (−2.32; −0.11) | 87.0 | (74.1; 93.5) | 2.23 | |
| **Diagnosis of the sample** | | | | | | | |
| Depression | 15 | −0.65 | (−1.01; −0.28) | 85.9 | (78.3; 90.8) | 3.94 | 0.164 |
| Anxiety | 7 | −1.18 | (−2.03; −0.34) | 84.1 | (68.9; 91.9) | 2.28 | |
| **Target group** | | | | | | | |
| Adults in general | 12 | −0.98 | (−1.47; −0.5) | 83.2 | (72; 89.9) | 2.67 | 0.961 |
| Other target groups | 11 | −0.96 | (−1.69; −0.24) | 91.4 | (86.6; 94.5) | 2.72 | |

Abbreviations: CIs, confidence intervals; g, Hedges' g; $n_{comp}$, number of comparisons in the analysis; NNT, number needed to treat.

in meta-analyses of e-mental health trials that were mainly conducted in HICs, which range from $g = −0.52$ (Moshe et al., 2021) to $g = −0.80$ (Pauley et al., 2021) for depressive and anxiety symptoms, respectively. Such differences have been previously identified in the literature on psychotherapy trials (Cuijpers et al., 2018) and raise important considerations. Although the precise reasons for these observations are not yet fully understood, it is possible that the limited availability of resources in LMICs may impact the adequacy

of control conditions, potentially leading to larger observed effects. Moreover, the quality of existing trials in LMICs can influence the outcomes (Cuijpers et al., 2018). Nonetheless, further investigation and rigorous research are essential to provide clearer insights into the effectiveness of psychological interventions in LMICs compared to HICs.

The finding that CBT programmes appeared more effective than other internet-based psychotherapy types in LMICs can be attributed to several factors. CBT's evidence-based and structured approach aligns well with online delivery, making it easier to adapt and implement in settings with less experience in diverse psychotherapy types. Additionally, the structured and time-limited nature of CBT makes it well-suited for internet-based delivery, particularly in resource-constrained settings where brief interventions are more feasible. However, it is important to recognise that CBT has dominated the internet-based interventions field, both in HICs and LMICs and thus, further research is needed to evaluate novel strategies and explore potential differences among therapeutic approaches. Moreover, this finding is based on subgroup analyses; thus, it do not imply causality since other variables may cause the differential outcomes between CBT and non-CBT interventions.

We did not find indications that the effects of digital interventions differ across several subgroup analyses, suggesting that these interventions may be effective despite the delivery format or whether they are culturally adapted or not. These results are in line with previous literature findings showing that guided and unguided interventions do not differ significantly in reducing symptoms of depression or anxiety (Moshe et al., 2021; Pauley et al., 2021). However, it should be noted that individual patient differences might lead to differential treatment effects. For instance, it has been found that guided interventions are more effective for individuals with more severe depressive symptoms compared to unguided interventions (Karyotaki, Efthimiou et al., 2021). Therefore, we cannot rule out the possibility that in LMICs, the provision of therapeutic guidance might lead to greater effects for some patient subgroups. Further, the current findings follow previous meta-analytic research indicating that the effects of psychological interventions do not differ significantly depending on cultural adaptation to the local context (Cuijpers et al., 2018). It should be noted, though, that the method of cultural adaptation was minimally reported across the included studies, suggesting that the actual extent of cultural adaptation might have an effect on digital interventions' efficacy, as reported before (Shehadeh et al., 2016).

To the best of our knowledge, no previous meta-analysis has examined the effects of digital interventions on quality of life. We found a small but significant effect ($g$ = 0.32) in improving the quality of life at the post-test, which was sustained over the longer term. This finding is in line with previous meta-analyses on psychotherapies that have found almost identical effects on quality of life ($g$ = 0.33) (Kolovos et al., 2016). Still, we should acknowledge that quality of life was examined only by seven studies, and future research should further explore quality-of-life outcomes.

Several limitations should be acknowledged. First, there was substantial heterogeneity across all analyses that remained unexplained in subgroup and meta-regression analyses. Such heterogeneity, however, is very common in meta-analyses focussing on psychological interventions in LMICs (Cuijpers et al., 2018; Karyotaki et al., 2022). The significant heterogeneity observed in meta-analyses of studies conducted in LMICs may arise from a range of factors. Diverse cultural, social and economic contexts can lead to variations in study populations and healthcare infrastructure, while differences in cultural beliefs and practices may influence treatment

outcomes. Moreover, variations in intervention protocols, cultural and delivery modes could contribute to the observed heterogeneity. Although excluding outliers in the current analysis led to a reduction in heterogeneity to a moderate level, no apparent differences were found in the identified outlier studies compared to the rest of the studies, leaving the specific reasons for the remaining heterogeneity uncertain.

Second, there was an indication of publication bias; thereby, the present outcomes should be interpreted cautiously. Nevertheless, the vast majority of analyses resulted in a smaller but significant effect after adjusting for publication bias, suggesting that the magnitude of digital interventions' true effect would probably be in the moderate range. Third, the overall risk of bias ranged from some concerns to high risk across all included studies, meaning that high-quality studies are needed to draw robust conclusions regarding the effects of digital interventions in LMICs. Fourth, although we did not find any significant differences between the countries' income classifications, we only identified studies from lower-middle or upper-middle countries. Therefore, more studies are needed to explore the effects of digital interventions in low-income countries. Fourth, although we found no significant differences between the countries' income classifications, we only identified studies from lower-middle or upper-middle countries. Notably, we did not identify any study meeting our eligibility criteria from Africa, a continent with a substantial and rapidly growing population. In many low-income countries, this lack of eligible RCTs may be attributed to systematic and structural challenges, such as limited access to electricity, mobile phones, and the internet, which hinder the development and research of digital interventions in low-income countries. Additionally, global and economic factors may impact research funding, infrastructure and collaboration opportunities, contributing to this disparity. To address this gap, researchers and policymakers should proactively tackle these challenges and prioritise evaluating digital interventions in low-income countries, especially in Africa, considering its rapidly growing population and potential to benefit from these therapeutic approaches.

Finally, we should acknowledge that our subgroup analysis results should be interpreted cautiously for several reasons. Firstly, it is important to note that finding a subgroup difference does not necessarily imply a direct causal relationship between the variable and the observed effect difference. Study characteristics may be confounded by various factors, such as differences in patient recruitment or varying levels of bias, which could impact the outcomes. Furthermore, the statistical power of subgroup analyses to detect differences relies on having an adequate sample size, and this power is limited when there are few studies or numerous subgroups (or both). Therefore, it is crucial to remember that the absence of evidence is not evidence of absence. It is possible that subgroups still exhibit different effects, but the lack of statistical power might have hindered their detection.

Despite the limitations, the present outcomes offer great possibilities for improving the access and availability of evidence-based psychological care through digital means in LMICs. We found that both guided and unguided interventions effectively reduce symptoms of depression and anxiety, suggesting that unguided interventions are also a good and probably cost-effective choice if there are limited resources for providing guidance. Next, digital interventions appear effective regardless of the target or age group, meaning they can be widely provided to cover the needs of people with symptoms of common mental disorders in LMICs. Nonetheless, based on the current findings, CBT interventions should be

preferred as they lead to larger effects. Overall, the present findings testify to the generalisability of the effectiveness of digital interventions for depression and anxiety in LMICs. These innovative interventions should be scaled up to reduce the impact of common mental disorders and hopefully contribute to bridging the gap between treatment supply and demand in low-resourced countries.

Abbreviations: CIs, confidence intervals; *g*, Hedges' *g*; *k*, number of comparisons in the analysis; NNT, number needed to treat; PI, prediction interval.

**Open peer review.** To view the open peer review materials for this article, please visit http://doi.org/10.1017/gmh.2023.50.

**Supplementary material.** The supplementary material for this article can be found at https://doi.org/10.1017/gmh.2023.50.

**Data availability statement.** Data are available upon request to the corresponding author, E.K. (e.karyotaki@vu.nl).

**Author contribution.** E.K. and C.M. drafted the manuscript. E.K., C.M. and M.H. performed the analysis. All the authors provided input into the study design and helped with manuscript writing. C.M. and O.P. contributed to the data extraction and helped with data synthesis and administration. R.A., V.P. and P.C. supervised the overall conduct of the study. All the authors read and approved the final manuscript.

**Financial support.** E.K. is supported by Nederlandse Organisatie voor Wetenschappelijk Onderzoek Veni (Grant No. VI.Veni.201G.053).

**Competing interest.** The authors declare no competing interests exist.

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
