## [Reviewer Report]

To the editor of the Global Mental Health,

Dear Editor, 

I am enclosing a submission to the Global Mental Health entitled ‘Digital Interventions for Common Mental Disorders in Low- and Middle-Income Countries: A Systematic Review and Meta-Analysis.

As you are undoubtedly aware, the provision of psychological interventions for common mental disorders is limited in Low- and Middle-Income Countries (LMICs) due to scarce healthcare resources. Digital interventions present a promising approach to addressing the enormous mental health gap in low-resourced settings. During the last few years, an increasing number of trials have been conducted examining the effects of digital interventions for individuals with depression and anxiety in LMICs. We conducted a systematic review and meta-analyses to examine the effects of these trials and provide the latest evidence in this field. 

In sum, we found that digital interventions are effective in reducing symptoms of depression and anxiety as well as improving quality of life over the short and the longer term. Most of the examined subgroup analyses (e.g., guided vs unguided digital interventions) did not reveal any significant differences. These results are very encouraging for the upscaling of digital interventions in LMICs. The manuscript is original, not previously published, and not under consideration elsewhere. All authors have declared a possible conflict of interest and have approved the attached version of the manuscript.

Sincerely, 

Eirini Karyotaki, PhD.

Associate Professor of Clinical Psychology

Department of Clinical, Neuro- and Developmental Psychology, VU University Amsterdam

Van der Boechorststraat 7, 1081 BT Amsterdam, the Netherlands

---

## [Reviewer Report]

This is a robust and well written paper that I enjoyed reviewing. Some comments and questions:

Introduction

• Contextualize the research more in terms of what has happened in HIC with CMDs and digital – clarify the digital boom is in HICs and what is filtering to LMICs?

• More background on the different contexts of HICs and LMIC e.g. smartphone penetration in LMIC may be growing but there is still a digital divide in lowest socioeconomic status groups/vulnerable groups, other contextual issues e.g. access to data – can also reflect on this in the discussion

• Can you include more up to date references on the burden of CMDs in LMICs?

Methods

• Was any guidance on reporting systematic reviews used e.g. PRISMA?

• The methods section would be strengthened by addition of some references for methods – e.g. for reasoning for using Hedges g as opposed to another measure (group size differences?); reference for intra-study correlation coefficient of ρ = 0.5. 

• Outline in the methods justification and reference for removing outliers from analysis

• What was done for missing outcome data – references for approach and justification

• Can you discuss the decision to include all age groups in the meta analysis – why was this deemed appropriate given that the intervention content may be quite different eg for adolescents?

• Clarify what the reader is to take from the subgroup analyses with respect to power to detect differences in subgroup analysis. 

Results

• Results are well presented

Discussion

• Can you discuss more on the finding that CBT programmes appeared to be more effective than other types of therapy – ideas on why? What do we know from HIC and how could this relate to LMIC

• You state that psychological interventions are more effective in LMICs – what is the reasoning and support for this? Discuss in relation to the view that some Western style approaches may actually be less appropriate and cultural adaptation may be minimal? What should the reader take away as to why psychological interventions could be more effective in LMICs?

• Limitations- heterogeneity of studies, can you say more about this – why is it common or expected in these types of meta analysis and what does this say about this meta analysis overall? Can you give more information on outliers – which studies, from which countries, any similarities between studies that were outliers? Is there anything to be taken away from looking at these outliers?

---

## [Reviewer Report]

Thank you for the opportunity to review this timely and important study that details how “digital interventions, with or without guidance, may effectively bridge the gap between treatment supply and demand in LMICs.” In addition, I agree with the conclusion that more research is needed to “draw firm conclusions regarding the magnitude of the effects of digital interventions.”

My major comment is in response to Question 1 posed by Cambridge in the review form: “For global reviews, how well does the review cover global content in the inclusion of research, presentation of results, and/or in the discussion and implications? And how could this be improved/expanded?”

As the study title states this is a review of digital interventions in low and middle-income countries. However, the authors note in the limitations few low-income countries were included. Some discussion of why so little research on digital interventions has been conducted in low-income countries is needed, as are the implications of this lack of research and directions for future study. I commend the authors on the inclusion of studies from LMICs including Brazil, China, Colombia, India, Indonesia, Iran, Lebanon, Pakistan, Peru, and Romania. Yet, there is not one study, nor is there even mention of the continent of Africa in this review. Authors should consider the implications of omitting a continent with one of the fastest growing populations that is currently home to more than a quarter of the world’s population. Attention should be paid to systematic and structural issues that make digital interventions in Africa challenging and thus under-researched and under-developed. For example, the cost of and inequitable access to electricity and mobile phones and the Internet (as our team Kreniske et al., 2021 JMIR; and other perhaps see G Porter and colleagues?). What are other global and economic factors that might contribute to this disparity? In short, what does it mean when digital interventions are not being evaluated in Africa?

Also, I would add in fact there are mental health digital interventions in Africa, but they have not yet been published or the randomized trials are not complete, including full disclosure my own small study. As well as a larger study, for instance, see Wainberg ML, et al., Technology and implementation science to forge the future of evidence-based psychotherapies: the PRIDE scale-up study. Evid Based Ment Health. 2021 Feb;24(1):19-24. doi: 10.1136/ebmental-2020-300199. Epub 2020 Nov 11. PMID: 33177149; PMCID: PMC8025148. A related team has published on a digital intervention to improve alcohol treatment (O’Grady et al., 2022), although alcohol use may be outside the scope of this manuscript. In addition, I am aware of partners such as StrongMinds in Uganda and Zambia who were in the process of publishing their work describing the provision of telehealth mental health care. Again, I am not sure these would qualify for the current review as the StrongMinds reports may not be RCTs. However, this in itself exposes concerns about how we as scientists assess, analyze and prioritize interventions for funding and scale-up. I do think RCTs are important but perhaps something to note or speak to in the manuscript as it relates low-income countries and the state of the evidence for digital interventions for mental health and Africa at the current moment and why the research in Africa is not at the same stage of development as other LMIC regions? Can this issue be addressed?

The following quote also highlights a major issue related to the omission of any African studies:

“Nevertheless, it should be noted that the method of cultural adaptation was minimally reported across the included studies, suggesting that the actual extend of cultural adaptation might have an effect on digital interventions’ efficacy as reported before (Shehadeh, Heim et al. 2016).” Africa is of course very diverse, even within country, but the lack of RCTs and culturally appropriate intervention evaluations is clearly related to the other issues raised in my review.

Minor: Authors use the word “nevertheless” 6 times. Consider removing or alternate phrase?

---

## [Reviewer Report]

Dear editor, 

We thank the reviewers for their valuable comments and suggestions. We have prepared a rebuttal with our responses to the reviewers' comments and submitted the revised version of our manuscript for your consideration. 

Sincerely, 

On behalf of the co-authors, 

Eirini Karyotaki